# Comparison of Proclarix, PSA Density and MRI-ERSPC Risk Calculator to Select Patients for Prostate Biopsy after mpMRI

**DOI:** 10.3390/cancers14112702

**Published:** 2022-05-30

**Authors:** Miriam Campistol, Juan Morote, Marina Triquell, Lucas Regis, Ana Celma, Inés de Torres, María E. Semidey, Richard Mast, Anna Santamaría, Jacques Planas, Enrique Trilla

**Affiliations:** 1Department of Urology, Vall d’Hebron Hospital, 08035 Barcelona, Spain; jmorote@vhebron.net (J.M.); mtriquell@vhebron.net (M.T.); lregis@vhebron.net (L.R.); acelma@vhebron.net (A.C.); jplanas@vhebron.net (J.P.); etrilla@vhebron.net (E.T.); 2Prostate Cancer Research Group, Vall d’Hebron, Research Institute, 08035 Barcelona, Spain; itorres@vhebron.net (I.d.T.); mesemidey@vhebron.net (M.E.S.); anna.santamaria@vhir.org (A.S.); 3Department of Surgery, Universitat Autònoma de Barcelona, 08193 Barcelona, Spain; 4Department of Pathology, Vall d’Hebron Hospital, 08035 Barcelona, Spain; 5Department of Morphological Sciences, Universitat Autònoma de Barcelona, 08193 Barcelona, Spain; 6Department of Radiology, Vall d’Hebron Hospital, 08035 Barcelona, Spain; rmast@vhebron.net

**Keywords:** clinically significant prostate cancer, PSA density, Proclarix, MRI-ERSPC, magnetic resonance imaging

## Abstract

**Simple Summary:**

The selection of proper candidates for prostate biopsy after magnetic resonance imaging (MRI) has usually been studied in the overall population with suspected prostate cancer (PCa). However, the performance of these tools can change regarding the Prostate Imaging-Reporting and Data System (PI-RADS) categories. We compared three different tools: PSA density, MRI-ERSPC risk calculator and Proclarix in 567 men with suspected PCa (PSA > 3 ng/mL and/or abnormal rectal examination) in one academic institution. All patients underwent multiple transrectal ultrasound guided biopsies after a multiparametric MRI was performed. We concluded that in the overall population, MRI-ERSPC RC outperformed PSA density and Proclarix, whereas in patients with lesions PI-RADS < 3 Proclarix was better than the other tools. However, no tool guaranteed 100% detection of clinically significant PCa in PI-RADS 4 and 5.

**Abstract:**

Tools to properly select candidates for prostate biopsy after magnetic resonance imaging (MRI) have usually been analyzed in overall populations with suspected prostate cancer (PCa). However, the performance of these tools can change regarding the Prostate Imaging-Reporting and Data System (PI-RADS) categories due to the different incidence of clinically significant PCa (csPCa). The objective of the study was to analyze PSA density (PSAD), MRI-ERSPC risk calculator (RC), and Proclarix to properly select candidates for prostate biopsy regarding PI-RADS categories. We performed a head-to-head analysis of 567 men with suspected PCa, PSA > 3 ng/mL and/or abnormal rectal examination, in whom two to four core transrectal ultrasound (TRUS) guided biopsies to PI-RADS ≥ three lesions and/or 12-core TRUS systematic biopsies were performed after 3-tesla mpMRI between January 2018 and March 2020 in one academic institution. The overall detection of csPCa was 40.9% (6% in PI-RADS < 3, 14.8% in PI-RADS 3, 55.3% in PI-RADS 4, and 88.9% in PI-RADS 5). MRI-ERSPC model exhibited a net benefit over PSAD and Proclarix in the overall population. Proclarix outperformed PSAD and MRI-ERSPC RC in PI-RADS ≤ 3. PSAD outperformed MRI-ESRPC RC and Proclarix in PI-RADS > 3, although none of them exhibited 100% sensitivity for csPCa in this setting. Therefore, tools to properly select candidates for prostate biopsy after MRI must be analyzed regarding the PI-RADS categories. While MRI-ERSPC RC outperformed PSAD and Proclarix in the overall population, Proclarix outperformed in PI-RADS ≤ 3, and no tool guaranteed 100% detection of csPCa in PI-RADS 4 and 5.

## 1. Introduction

Early detection of clinically significant prostate cancer (csPCa) decreases the specific mortality of PCa [1]. The classic diagnostic approach to PCa, based on systematic biopsies after elevated serum prostate-specific antigen (PSA) and/or abnormal digital rectal examination (DRE) [2], has been disapproved due to high rates of unnecessary biopsies and an over detection of insignificant PCa (iPCa) [3]. Recent improvements in early detection of csPCa come from multiparametric magnetic resonance imaging (mpMRI) and guided biopsies [2]. 

The efficacy of this new diagnostic strategy for csPCa can still be improved by a proper selection of candidates for prostate biopsy, particularly in uncertain cases [3]. The current negative predictive value (NPV) of mpMRI, when the Prostate Imaging-Reporting and Data System (PI-RADS) category is below 3, reaches 95% [4,5], and that is why most clinicians recommend avoiding biopsies in these cases. Conversely, men with PI-RADS categories greater than 3 have a likelihood of csPCa ranging from 55% to 95% [6], and almost all clinicians recommend scheduling prostate biopsies in these circumstances. The probability of csPCa in PI-RADS category 3 is not higher than 20%, making this an uncertain case [7,8].

Among the proposed tools for improving the proper selection of candidates for prostate biopsy after mpMRI, PSA density (PSAD) has recently emerged, as MRI provides the most accurate measurement of prostate volume without additional cost [9]. PSAD has been analyzed according to PI-RADS categories and different thresholds have been proposed depending on the results [10]. Currently, there is no ideal marker to use after mpMRI [11]; the new marker Proclarix might be an adequate candidate due to its high sensitivity for csPCa but has not yet been analyzed by PI-RADS category [12,13]. This test has recently been introduced, providing a risk score of csPCa from 1 to 100% with a cut-off at 10%, obtaining a high sensitivity and a high negative predictive value (90 and 95% respectively) [14]. It is based on the serum determination of Thrombospondin-1, Cathepsin D, PSA and % fPSA, together with age. In addition, predictive models are attractive tools when they incorporate easily assessed clinical variables, when they are externally validated, and when web or smartphone applications (apps) are provided for their easy use in clinical practice, as in the case of the recent MRI-ERSPC risk calculator (RC). However, none of these has been evaluated by the PI-RADS category [15]. 

Since the incidence of csPCa increases by PI-RADS category, we hypothesized that changes in the predictive value of tools to improve the proper selection of candidates for prostate biopsy will be expected [10]. Therefore, we primarily seek to change the evaluation paradigm of these tools after verifying our hypothesis. We analyzed the usefulness of PSAD, the MRI-ERSPC RC, and the new marker Proclarix in a population of men with suspected PCa, as well as evaluating them by PI-RADS category.

## 2. Materials and Methods

### 2.1. Design, Setting and Participants

This is a prospective comparative study between PSAD, MRI-ERSPC RC and Proclarix in 567 consecutive men with suspected PCa due to PSA levels > 3 ng/mL and/or abnormal DRE scheduled for a 3-tesla mpMRI prior to biopsy from 15 January 2018 to 20 March 2020 in one academic institution. Twelve core transrectal ultrasound (TRUS) systematic-biopsies were performed in all participants, and two to four core TRUS cognitive fusion biopsies were taken in those patients with suspicious lesions (PI-RADSv.2 ≥ 3). Men with PCa on active surveillance and those with symptomatic benign prostatic hyperplasia treated with 5-α-reductase inhibitors were excluded. This project was approved by the Institutional Ethics Committee (PR-AG129/2020), and informed consent was obtained from study participants.

### 2.2. Intervention

Proclarix was assessed from serum samples obtained just before prostate biopsies were performed and stored at −80 °C (Collection 0003439; https://biobancos.isciii.es (accessed on 13 December 2021)). Thrombospondin 1 (THBS-1), Cathepsin D (CTD), total PSA, and free PSA were determined at Proteomedix (Zurich-Schlieren, Switzerland). THBS-1 and CTD levels were measured with specific immunoassays described previously [16]. Total PSA and free PSA were analyzed for all samples with the Roche Cobas immunoassay system (Roche Diagnostics, Rotkreuz, Switzerland), and age was calculated using an algorithm that reported a score ranging from 0% to 100% [14,17]. PSAD (ng/mL/cc) was estimated from the PSA level determined in the Proclarix assessment and the prostate volume reported in the pre-biopsy mpMRI. The MRI-ERSPC likelihood of high-grade PCa (Gleason ≥ 3 + 4) was estimated for every man through the SWOP web application (Prostate Cancer Research Foundation, Reeuwijk) at www.prostatecancer-riskcalculator.com (accessed on 21 January 2022) [18]. The MRI-ERSP RC includes serum PSA (0.5 to 50 ng/mL), repeat biopsy (yes/no), DRE (normal/abnormal), prostate volume (10–110 cc), age (50–75 years), and PI-RADSv.1 [15]. For these calculations, the MRI-based prostate volume was introduced as well as the PI-RADSv.2 categories [19]. When the observed values were not within the accepted range, the closest minimum or maximum accepted value was entered. 

### 2.3. Endpoint Measurements

The CsPCa detection rate and avoidable prostate biopsies were the primary endpoint measurements. CsPCa was confirmed when the ISUP (International Society of Uropathology) grade group was ≥2 [20,21].

### 2.4. Statistical Analyses

The association between quantitative variables was assessed with the Mann–Whitney U test and the Kruskal–Wallis test. The associations between qualitative variables were analyzed with a Chi-square test. The odds ratios (OR) and 95% confidence interval (95% CI) were also estimated. Receiver operating characteristic (ROC) curves were constructed and areas under the curve (AUC) were estimated and compared with the DeLong test [22,23]. A decision curve analysis (DCA) was performed to study the net benefits [24] and clinical utility curves (CUC) were generated to assess the difference between missed csPCa and avoided biopsies across the continuous likelihood of csPCa [25]. The performance of predictors with the selected thresholds were analyzed based on sensitivity, specificity, positive and negative predictive values (PPV and NPV), accuracy, rates of avoided biopsies, and rate of missed csPCa. A *p*-value of less than 5% was considered significant. SPSS v.25 (IBM Corp., Armonk, NY, USA) and R programming language v.3.3.1 (The R Statistical Foundation, Vienna, Austria) were used.

## 3. Results

### 3.1. Characteristics of the Study Population and Distribution of Overall PCa, csPCa, and iPCa by PI-RADS Category

The characteristics of the study population are summarized in Table 1. We highlight a median age of 69 years and a PSA of 7.0 ng/mL. In addition, 19.2% of patients had an abnormal DRE, 23.5% were repeated biopsies, and 8.6% had a family history of PCa. The distribution by PI-RADS categories was 17.6% with PI-RADS < 3, 29.8% with PI-RADS 3, 33.5% with PI-RADS 4, and 19% with PI-RADS 5. The overall rate of detected PCa was 52.6%, 40.9% of csPCa, and 11.7% of iPCa. CsPCa was detected in 6% of men with PI-RADS < 3, 14.8% in PI-RADS 3, 55.3% in PI-RADS 4, and 88.9% in PI-RADS 5, *p* < 0.001.

### 3.2. Overall Efficacy, Net Benefit, and Clinical Utility of mpMRI, PSAD, MRI-ERSPC RC, and Proclarix, and Overall Performances after the Selection of Appropriate Thresholds

ROC curves analyzing the efficacy of mpMRI, PSAD MRI-ERSPC RC, and Proclarix for the detection of in the overall population study are presented in Figure 1a. MRI-ERSPC RC showed an AUC of 0.856 (95% CI: 0.824–0.888); mpMRI, 0.831 (95% CI: 0.705–0.786); Proclarix, 0.745 (95% CI: 0.705–0.786); and PSAD, 0.740 (95% CI: 0.698–0.782), with *p* = 0.038. DCAs showed the highest net benefit for mpMRI at threshold probabilities between 0.1 and 0.45, while MRI-ERSPC RC when the threshold probability was higher, Figure 1b. CUCs showed the largest area between csPCa missed and avoided biopsy rates at all threshold probabilities, as shown in Figure 1c. 

Based on the highest possible sensitivity for csPCa, the selected threshold for mpMRI was PI-RADS 2, with 10% for Proclarix, 0.07 ng/mL/cc for PSAD, and 3% for MRI-ERSPC RC. The performances of these tools based on the selected thresholds are summarized in Table 2. We note that mpMRI exhibited a sensitivity of 97.4%, avoiding 17.6% of prostate biopsies. These parameters were 97.4% and 16.8%, respectively, for Proclarix, 90.1% and 21.0% for PSAD, and 94.4% and 20.6% for MRI-ERSPC RC. The NPVs were 94%, 93.7%, 80.7%, and 88.9%, respectively. The Grade Group (GG) of missed csPCa for each tool are also summarized in Table 2. 

### 3.3. Efficacy, Net Benefit, Clinical Utility, and Performance of PSAD, MRI-ERSPC RC, and Proclarix by PI-RADS Category 

We will now analyze the behavior of MRI-ERSPC RC, PSAD, and Proclarix for the detection of by PI-RADS category using the previously selected thresholds. ROC curves and the AUCs for every tool are presented in Figure 2. We note different morphologies of these curves and AUCs by PI-RADS categories and in those observed in the overall population. The AUCs of MRI-ERSPC RC in men with PI-RADS < 3 was 0.516 (95% CI: 0.338–0.693), Figure 2a; 0.657 (95% CI: 0.547–0.766) in men with PI-RADS 3, Figure 2b; 0.676 (95% CI: 0.601–0.752) in men with PI-RADS 4, Figure 2c; and 0.765 (95% CI: 0.605–0.926) in men with PI-RADS 5, Figure 2d, with *p* = 0.031. We found that the largest AUC in men with PI-RADS ≤ 3 was for Proclarix, at 0.610 (95% CI: 0.416–0.803) in men with PI-RADS < 3, Figure 2a and 0.703 (95% CI: 0.620–0.786) in those with PI-RADS 3, Figure 2b, with *p* = 0.039. In contrast, PSAD exhibited the highest AUC in men with PI-RADS >3, at 0.704 (95% CI: 0.631–0.777) in men with PI-RADS 4, Figure 2c and 0.826 (95% CI: 0.706–0.945) in those with PI-RADS 5, Figure 2d, with *p* = 0.028. DCAs by PI-RADS category showed a net benefit of Proclarix over PSAD and MIR-ERSPC RC in men with PI-RADS ≤ 3, especially at low threshold probabilities of csPCa, while neither tool exhibited a clear net benefit in men with PI-RADS 4 and 5. The CUCs by PI-RADS category are shown in Figure 3a–d. We noted that the area between the rates of avoided biopsies and missed csPCa was greater for Proclarix in men with PI-RADS ≤ 3 and for PSAD in men with PI-RADS > 3. 

Proclarix, PSAD, and MRI-ERSPC RC using the selected thresholds with the highest sensitivity for csPCa by PI-RADS category are summarized in Table 3. We found that Proclarix was able to detect 100% of csPCa in men with negative mpMRI and men with PI-RADS 3, avoiding 30% and 21.3% of prostate biopsies, respectively. Proclarix was also able to reduce 12.1% of prostate biopsies in men with PI-RADS 4 but misdiagnosed 4.8% of csPCa; these rates were 5.6% and 1%, respectively, in men with PI-RADS 5. PSAD was able to avoid between 29% and 9.3% of prostate biopsies by PI-RADS categories but missed between 50% and 4.2% of csPCa, respectively. MRI-ERSPC RC was able to avoid between 63% and 0% of prostate biopsies by PI-RADS categories but missed between 83.3% and 0% of csPCa, respectively. The GG distribution of misdiagnosed csPCa by PI-RADS category for each tool is also shown in Table 3.

## 4. Discussion

This is the first head-to-head study between PSAD, the externally validated MRI-ERSPC predictive model, and the new marker Proclarix for the proper selection of candidates for prostate biopsy after mpMRI. Morote et al. [26] analyzed the behavior of Proclarix in the same series of patients but in the subgroup of men with PI-RADS 3 category. The results obtained demonstrated that Proclarix outperformed PSAD in the detection of csPCa in this specific scenario (PI-RADS 3 category), considered the most uncertain of PI-RADS. This present work incorporated an evaluation of Proclarix in all PI-RADS categories compared to PSAD and the predictive model. New and clinically relevant information is provided from the analysis of these tools by PI-RADS category, in addition to the study carried out in the overall population of men with suspected PCa, which has been the most frequent method to report their performances. Clinicians need to know when, where, and how they should use these tools to avoid unnecessary prostate biopsies in exchange for acceptable rates of failure to detect csPCa. The relevant questions that clinicians ask are: (1) How many biopsies are we willing to perform to improve the current negative predictive value of mpMRI?; (2) What is the csPCa loss rate that we are willing to accept by PI-RADS category?; and (3) At what cost? For these purposes, we demonstrate that analysis by PI-RADS category is required.

When the entire population of men with suspected PCa was analyzed, the MRI-ERSPC model was the most efficient tool for csPCa detection. In addition, the MRI-ERSPC model exhibited a net benefit over PSAD and Proclarix and outperformed both according to the differential area between rates of avoided biopsies and missed csPCa shown in the CUCs. However, the performance of these tools changed when analyzed by the PI-RADS category. The new marker Proclarix, which is very sensitive for csPCa [12,13], was the most efficient and clinically useful tool in men with PI-RADS ≤ 3. Proclarix increased the negative predictive value of mpMRI from 94% to 100%, while a prostate biopsy was required in 70% of men with negative mpMRI [4,5]. In contrast, PSAD recommended prostate biopsy in 69% of men with negative mpMRI, leaving 50% of csPCa undetected [10,26]. Finally, MRI-ERSPC RC recommended biopsy in 37% of men with negative mpMRI but missed 83.3% of csPCa. In men with the challenging PI-RADS category 3, Proclarix was also the most efficient tool, and exhibited 100% sensitivity for csPCa while avoiding 21.3% of prostate biopsies. PSAD would avoid 26.2% of prostate biopsies but would miss 16% of csPCa. MRI-ERSPC RC would avoid 29.6% of prostate biopsies but would also miss 16% of csPCa. In men with PI-RADS 4, PSAD was the most efficient tool, avoiding 18.4% of prostate biopsies but missing 11.4% of csPCa. Proclarix would avoid 12.1% of prostate biopsies but would miss 4.8% of csPCa. MRI-ERSPC RC would avoid 9.3% of prostate biopsies and would miss 4.2% of csPCa. Finally, in men with PI-RADS 5, in whom 88.9% of csPCa was detected, PSAD was also the most efficient tool, avoiding 9.3% of prostate biopsies while missing 4.2% of csPCa. Proclarix would avoid 5.6% of prostate biopsies and would miss 1% of csPCa. MRI-ERSPC RC would not avoid any prostate biopsies. The PI-RADS > 3 are categories with high and very high-risk of csPCa in addition to an increased aggressiveness [27,28,29]. Therefore, clinicians are unwilling to miss any csPCa to avoid some prostate biopsies; therefore, only tools that guarantee 100% sensitivity for csPCa are acceptable in this category. 

This study has some limitations. Although 567 men with suspected PCa was a sizeable cohort and there was an accurate representation of the incidence of the PI-RADS category, the low cases of csPCa in men with negative mpMRI and PI-RADS category 3 is a limitation. MRI-ERSP RC was designed to use PI-RADS v.1 in men up to 75 years old with serum PSA up to 20 ng/mL and prostate volume up to 110 ccs; however, we used PI-RADS v.2 and did not limit age or prostate volumes. Additionally, since it is a prospective study in a single center, the risk of bias could be higher. An external and multicenter analysis should be performed. Finally, although the used definition of csPCa in prostate biopsies is widespread, it does not represent the true pathology observed in surgical specimens. 

## 5. Conclusions

This study suggests a change in the paradigm of evaluating tools for the proper selection of candidates for prostate biopsy after mpMRI. Evaluations in the entire population of men with suspected PCa are insufficient. We suggest that evaluations of these tools regarding PI-RADS categories are needed to provide clinicians with sufficient and useful information to meet their expectations for the early detection of csPCa. MRI-ERSPC RC, was the most effective tool for the adequate selection of candidates for prostate biopsy when the entire population was analyzed. However, Proclarix was the most useful in men with PI-RADS ≤ 3. None of the tools exhibited the 100% sensitivity desired for csPCa in high and very high-risk PI-RADS categories. Taking into consideration the results of this study, Proclarix seems to be a relevant tool. It is especially useful in those men with PI-RADS ≤ 3 lesions in the mpMRI to decide whether to biopsy the patient. 

## Figures and Tables

**Figure 1 cancers-14-02702-f001:**
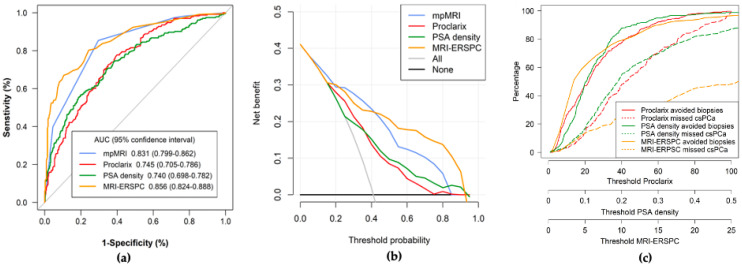
Efficacy (**a**), net benefit (**b**) and clinical utility (**c**) of Proclarix, PSAD and MRI-ERSPC model for csPCa detection in the overall population.

**Figure 2 cancers-14-02702-f002:**
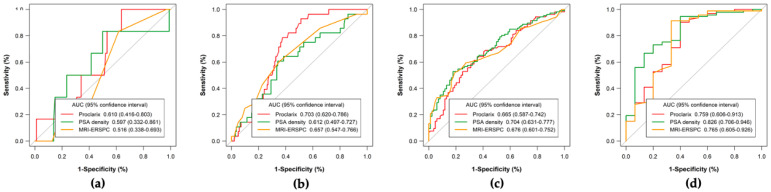
Efficacy of Proclarix, PSAD, and MRI-ERSPC model for csPCa detection regarding PI-RADS categories. PI-RADS < 3 (**a**), PI-RADS 3 (**b**), PI-RADS 4 (**c**), and PI-RADS 5 (**d**).

**Figure 3 cancers-14-02702-f003:**
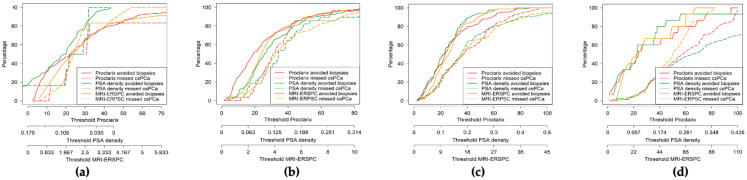
Clinical utility of Proclarix, PSAD, and MRI-ERSPC model for csPCa detection regarding PI-RADS categories. PI-RADS < 3 (**a**), PI-RADS 3 (**b**), PI-RADS 4 (**c**), and PI-RADS 5 (**d**).

**Table 1 cancers-14-02702-t001:** Characteristics of the study cohort.

Characteristic	Measurement
Number of cases	567
Median age, years (IQR)	69 (63–74)
Median total PSA, ng/mL (IQR)	7.0 (4.9–11.2)
Abnormal DRE, *n* (%)	109 (19.2)
Median free PSA, ng/mL (IQR)	1.1 (0.7–1.7)
Median prostate volume, mL (IQR)	55 (40–76)
Median percent free PSA, % (IQR)	15.1 (10.7–20.6)
Median PSA density, ng/mL/cc (IQR)	0.13 (0.09–0.21)
Repeat biopsy, *n* (%)	133 (23.5)
Family history of PCa, *n* (%)	48 (8.6%)
PI-RADS, *n* (%)	
1–2	100 (17.6)
3	169 (29.8)
4	190 (33.5)
5	108 (19.0)
Overall PCa detection, *n* (%)	298 (52.6)
csPCa detection, *n* (%)	232 (40.9)
iPCa detection, *n* (%)	66 (11.7)

IQR = Interquartile range; PCa = Prostate Cancer; csPCa = clinically significant PCa; iPCa = insignificant PCa.

**Table 2 cancers-14-02702-t002:** Overall performance of mpMRI, Proclarix, PSAD, and MRI-ERSPC model for csPCa detection.

Parameter	mpMRI	Proclarix	PSAD	MRI-ERSPC
Cut-off	1–2 PI-RADS	10%	0.07 ng/mL/cc	3%
Sensitivity (%)	226/232 (97.4)	226/232 (97.4)	209/232 (90.1)	219/232 (94.4)
Specificity (%)	94/335 (28.1)	89/335 (26.6)	96/335 (28.7)	104/335 (31.0)
Negative predictive value (%)	94/100 (94.0)	89/95 (93.7)	96/119 (80.7)	109/117 (88.9)
Positive predictive value (%)	226/467 (48.4)	226/472 (47.9)	209/448 (46.7)	219/450 (48.7)
Accuracy (%)	320/567 (56.4)	315/567 (55.6)	305/567 (53.8)	323/567 (57.3)
Avoidable biopsies	100/567 (17.6)	95/567 (16.8)	119/567 (21.0)	117/567 (20.6)
Misdiagnosis of csPCa (%)	6/232 (2.6)	6/232 (2.6)	23/232 (9.9)	13/232 (5.6)
GG2	4	3	10	8
GG3	1	2	6	1
GG4	1	1	4	2
GG5	0	0	3	0

mpMRI = multiparametric magnetic resonance imaging; PSAD = prostate-specific antigen density; csPCa = clinically significant prostate cancer; PI-RADS = prostate imaging-report and data system; GG = grade group.

**Table 3 cancers-14-02702-t003:** Characteristics of Proclarix, PSAD and MRI-ERSPC regarding PI-RADS category.

PI-RADS	Sensitivity	Specificity	NPV	PPV	Accuracy	Avoidable Biopsies	Misdiagnosis of csPCa	GG2	GG3	GG4	GG5
Proclarix (cut-off 10%)											
1–2	6/6(100)	30/94 (31.9)	30/30 (100)	6/70 (8.6)	36/100 (36)	30/100 (30)	0/6 (0)	0	0	0	0
3	25/25 (100)	36/144 (25.0)	36/36 (100)	25/133 (19.8)	61/169 (36.1)	36/169 (21.3)	0/25 (0)	0	0	0	0
4	100/105 (95.2)	18/85 (21.2)	18/23 (78.3)	100/167 (59.9)	118/190 (62.1)	23/190 (12.1)	5/105 (4.8)	2	1	1	1
5	95/96 (99.0)	5/12 (41.7)	5/6 (83.3)	95/102 (93.1)	100/108 (92.6)	6/108 (5.6)	1/96 (1.0)	1	0	0	0
PSAD (cut-off 0.07 ng/mL/cc)											
1–2	3/6 (50)	26/94 (27.7)	26/29 (89.7)	3/71 (4.2)	29/100 (29.0)	29/100 (29.0)	3/6 (50.0)	1	1	1	0
3	21/25 (84.0)	41/144 (28.5)	41/45 (91.1)	21/124 (16.0)	62/169 (36.7)	45/169 (26.2)	4/25 (16.0)	4	0	0	0
4	93/105 (88.6)	13/85 (27.1)	23/35 (65.7)	93/155 (60.0)	116/190 (61.1)	35/190 (18.4)	12/105 (11.4)	4	3	3	2
5	92/96 (95.8)	6/12 (50.0)	6/10 (60.0)	92/98 (93.9)	98/108 (90.7)	10/108 (9.3)	4/96 (4.2)	0	2	1	1
MRI-ERSPC model (cut-off 3%)											
1–2	1/6 (16.7)	58/94 (61.7)	58/63 (92.1)	1/37 (2.7)	59/100 (59)	63/100 (63)	5/6 (83.3)	3	1	1	0
3	21/25 (84)	46 (31.9)	46/50 (92)	21/119 (17.6)	67/169 (39.6)	50/169 (29.6)	4/25 (16)	4	0	0	0
4	103/105 (98.1)	2/86 (2.3)	2/4 (50)	103/186 (15.3)	104/190 (54.8)	4/190 (2.1)	2/105 (1.9)	1	1	0	2
5	96/99 (100)	NA	NA	96/108 (88.9)	96/108 (87.9)	0/108 (0)	0/96 (0)	0	0	0	0

PI-RADS = prostate imaging-report and data system; NPV = negative predictive value; PPV = positive predictive value; csPCa = clinically significant prostate cancer; GG = grade group.

## Data Availability

The data presented in this study are available on request from the corresponding author.

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
