# Peer review of "Comparison of Proclarix, PSA Density and MRI-ERSPC Risk Calculator to Select Patients for Prostate Biopsy after mpMRI"

_cancers, 2022, doi:10.3390/cancers14112702_

Round 1
Reviewer 1 Report
Dear Authors!
Nice and well done study - with an interessting topic/question: ´can we improve our recomendations for biopsy or not to biopsy the prostate after mp MRI of the prostate gland.'
it needs some minor revisions:
herby you can find my review:
- I would recommend to change the title in: "Comparison of Proclarix,....to select patients for prostate biopsy after mpMRI"...
- please number all your pages.
- Introduction: please explain/introduce "Proclarix" and the Proclarix risk score; "We analyzed the usefulness of PSAD, the MRI-ERSPC RC, and the new marker Proclarix in a population of men with suspected PCa, as well as evaluating them by PI-RADS category." I think you correlated these 3 tools with the resultes of the mpMRI (PI-RADS score) ?!
- Material and methods. It was a prospective study - wasn`t it - You should metion it. You performed a systematic random biopsy (12x) and a targeted biopsy of the Pi-RADS leason (2-4) - so you performed a cognitive fusion biopsy of the prostate.
- Results: o.k
- Discussion: I would add one additional "limitation": singel center study
- Conclusions: What are your conclusions? What is your take home message? Did your results change your way of biopsy? Maybe you recomend f.i. PI-RADS<3 performe an additional Proclarix test if you have also an abnormal DRE, or family history of PCa?, PI-RADS 3 perform always a Proclarix if...., and perform always prostate biopsy in PI-RADS 3, 4 and 5 - ?
kindly regards
Author Response
Thank you very much for your comments and your corrections. All of them have been taken into consideration and changed in the manuscript.
1. I would recommend changing the title in: "Comparison of Proclarix....to select patients for prostate biopsy after mpMRI"...
Ok. Thank you very much for the recommendation. In the actual lines 2-4 we changed the title to “Comparison of Proclarix, PSA density and MRI-ERSPC Risk Calculator to select patients for prostate biopsy after mpMRI”
2. Please number all your pages.
Thank you very much for the advice. We added numbers to all pages.
3. Introduction: please explain/introduce "Proclarix" and the Proclarix risk score; "We analyzed the usefulness of PSAD, the MRI-ERSPC RC, and the new marker Proclarix in a population of men with suspected PCa, as well as evaluating them by PI-RADS category." I think you correlated these 3 tools with the results of the mpMRI (PI-RADS score)?!
Yes. You are right that Proclarix risk score was not clear enough in the previous text. We added in the actual lines from 89 to 95: “This test has recently been introduced providing a risk score of csPCa from 1 to 100% with a cut-off at 10% obtaining a high sensitivity and a high negative predictive value (90 and 95% respectively) [17]. It is based on the serum determination of Thrombospondin-1, Cathepsin D, PSA and %fPSA, together with age.”
4. Material and methods. It was a prospective study - wasn`t it - You should mention it. You performed a systematic random biopsy (12x) and a targeted biopsy of the Pi-RADS lesson (2-4) - so you performed a cognitive fusion biopsy of the prostate.
Yes, it is a prospective study, and we did not specify that it in the article. Moreover, the biopsies were cognitive fusion. We added this information in the material and methos section, in the actual lines 168.
5. Results: o.k
6. Discussion: I would add one additional "limitation": single center study
Effectively, thank you very much. I also consider being a single center study a limitation of the study. Therefore, we added it in the actual line 392 to 394.
7. Conclusions: What are your conclusions? What is your take home message? Did your results change your way of biopsy? Maybe you recommend f.i. PI-RADS<3 perform an additional Proclarix test if you have also an abnormal DRE, or family history of PCa?, PI-RADS 3 perform always a Proclarix if...., and perform always prostate biopsy in PI-RADS 3, 4 and 5 - ?
Yes, you are right. We have added this some of our conclusions in the actual lines from 406 to 408: “Taking into consideration the results of this study, Proclarix seems to be a relevant tool. It is especially useful in those men with PI-RADS <3 lesions in the mpMRI to decide whether to biopsy the patient”.
Reviewer 2 Report
Authors should be congratulated for their great contribution to the challenging topic. All future perspectives should lead to improving prostate cancer detection and reducing investigations number. Moreover, the future purpose is to create new and better algorithms to properly manage early-stage PCa patients, avoiding overdiagnosis and overtreatment. Furthermore, a detailed pre-operative assessment of PCa is imperative both for the surgeons and for PCa patients’ expectations. The manuscript is well written, tables and figures are clear. The paper is suitable for publication.
Author Response
Thank you very much for your comment and you appreciation.
Reviewer 3 Report
The authors published a very similar article in March 2022. The authors come to the same conclusion. Proclarix test can optimize the detection of clinically significant prostate cancer in men with a score of 3 on the Prostate Imaging-Reporting and Data System for magnetic resonance imaging scans. So I could not feel anything new.
Eur Urol Open Sci. 2022 Mar; 37: 38–44. PMID: 35243388
Improving the Early Detection of Clinically Significant Prostate Cancer in Men in the Challenging Prostate Imaging-Reporting and Data System 3 Category
Other comments are Confirm grammar and content.
In Simple Summary
“The selection of proper candidates for prostate biopsy after magnetic resonance imaging (MRI) have usually been … “ should correct to “The selection of proper candidates for prostate biopsy after magnetic resonance imaging (MRI) has usually been …”
The title of table 3 should be described.
Figure 2 and Figure 3 are incomplete.
Author Response
Thank you for the appreciations. All comments have been taken into consideration. This is a study based on the same cases but introducing RSPC which has not been introduced previously. Moreover, we included a comparative study of the entire series regarding the PI-RADS caregories.
Reviewer 4 Report
Dear Authors,
I read with interest the entitled “Study of Proclarix, PSA density and MRI-ERSPC Risk Calculator to select patients for prostate biopsy” and I congratulate for the great effort behind it.
In this study the authors aimed to retrospectively evaluate the utility of a new biomarker (i.e., Proclarix), a risk calculator and PSAD, after MRI to select patients for prostate Bx in men with clinical suspicious. The main strength of the paper is that it analizes 3 tools, also stratifying results by PI-RADS category.
I found the present study interesting, well written and fluent to read, with interesting data. It concerns with an actual topic, which is constantly under debate and of major importance.
- The title is descriptive of what authors have explored in their work - one suggestion is to add “after MRI” - or something similar in order to focus on this concept -, as the main analyses of the paper have been carried out stratified by PI-RADS.
- The background and scientific rationale for carrying out the study are well presented. Due to the increasing importance of new biomarkers for csPCa detection - especially in association with MRI - and since there is not an ideal one, I would suggest mentioning also other available markers which have been explored with the same intent of this paper (e.g., doi: 10.1097/JU.0000000000001361; doi: 10.1007/s00345-020-03359-w).
- Tables and Figures are clear and not repetitive, as well as Results section.
- Statistical assessment is well conducted and the paper results methodologically correct. I have just some concerns and some aspects should be clarified: -which approach has been used to perform target Bx (e.g., cognitive or fusion?) That should be more clearly stated; - Similarly, it should be reported if there were a timeframe range between MRI and Bx (e.g., patients who had their MRI >12 months prior to Bx were excluded?); - it is not fully clear if PI-RADS v1 or 2 was used for the calculator (it was designed with version 1, yet authors used 2 - I suppose. Please be clearer on that, could be confusing.
- Discussion is adequately implemented with the relevant literature, and interpretations and conclusions are well stated and justified by results.
I have no other concerns or suggestions.
Author Response
1. The title is descriptive of what authors have explored in their work - one suggestion is to add “after MRI” - or something similar in order to focus on this concept -, as the main analyses of the paper have been carried out stratified by PI-RADS.
It is true that the entire study focuses on Proclarix, PSA density and MRI-ERSPC with the PI-RADS results. We added: after mpMRI in the title as suggested.
2. The background and scientific rationale for carrying out the study are well presented. Due to the increasing importance of new biomarkers for csPCa detection - especially in association with MRI - and since there is not an ideal one, I would suggest mentioning also other available markers which have been explored with the same intent of this paper (e.g., doi: 10.1097/JU.0000000000001361; doi: 10.1007/s00345-020-03359-w)
You are right. New liquid biomarkers have been developed and evaluated together with mpMRI for the detection of csPCa. We added: “Moreover, other biomarkers such as SelectMD, PCA3 or 4Kscore have been evaluated in association with mpMRI for the detection of csPCa” to the actual line 93 to 95.
Furthermore, we also added: “Other recent studies on new biomarkers have focused on how these diagnostic and prognostic tests act when related to mpMRI. Busseto et al. [29]evaluated the association of SelectMDx with mpMRI. However, the analysis of the PSAD together with mpMRI obtained a greater sensitivity compared to the association between SelectMDx and mpMRI (75.6% vs 80%). In addition, the combination of 4Kscore with mpMRI has also been evaluated avoiding 19.3% of prostate biopsies while missing 2.4% of csPCa [30].” in the actual lines 379 to 384.
3. Tables and Figures are clear and not repetitive, as well as Results section.
Thank you very much for your comment. We very much appreciate it.
4. Statistical assessment is well conducted and the paper results methodologically correct. I have just some concerns and some aspects should be clarified: -which approach has been used to perform target Bx (e.g., cognitive or fusion?) That should be more clearly stated; - Similarly, it should be reported if there were a timeframe range between MRI and Bx (e.g., patients who had their MRI >12 months prior to Bx were excluded?); - it is not fully clear if PI-RADS v1 or 2 was used for the calculator (it was designed with version 1, yet authors used 2 - I suppose. Please be clearer on that, could be confusing.
Yes. We agree that it is a little bit confusing. This is why we added in the actual lines 167-169: “2 to 3-core TRUS cognitive fusion biopsies were taken in those patients with suspicious lesions (PI-RADSv.2 >3).”
The predictive model (MRI-ERSP) was developed with a selection of patients in which the PI-RADS v1. was used. In our analysis PI-RADSv.2 was applied and introduced in the risk calculator. However, MRI-ERSP does not specify which one to use. It is one of the limitations of the study as it is mentioned in the actual lines from 387 to 390.
5. Discussion is adequately implemented with the relevant literature, and interpretations and conclusions are well stated and justified by results.
Thank you very much for your comment.
Round 2
Reviewer 3 Report
The manuscript is well written, tables and figures are clear. The paper is suitable for publication.
Author Response
The manuscript is well written, tables and figures are clear. The paper is suitable for publication.
Thank you very much for your comment.